**Data Availability Statement:** All relevant data are within the manuscript.

**Funding:** This work was supported by the Fondazione di Sardegna (U500.2013/AI.424.MGB to PSD) and by the University of Sassari Research

# Memantine effects on ingestion microstructure and the effect of administration time: A within-subject study

**Adriana Galistu, Paolo S. D'Aquila** *

Dipartimento di Scienze Biomediche, Università di Sassari, Sassari, Italy

* dsfpaolo@uniss.it

## Abstract

In a between-subject comparison of two memantine administration schedules we observed that treatment with the NMDA receptor antagonist memantine before testing sessions reduced ingestion of a 10% sucrose solution in rats, due to reduced licking burst size, thus suggesting a blunted hedonic response. Conversely, daily post-session administration reduced burst number, indicating a reduced level of behavioural activation, likely due to the development of conditioned taste aversion (CTA). In this study, the effect of pre-session and post-session memantine administration was investigated within-subjects. Memantine was administered in daily intraperitoneal injections for 13 days, on alternate days, either 1-h before–"before testing" sessions—or immediately after a 30-min session–"after testing" sessions. The effects on the microstructure of licking for a 10% sucrose solution were examined in the course of treatment and for 21 days after treatment discontinuation. The results show reduced burst size in the "before testing" sessions, without effects on the intra-burst lick rate, an index of motoric effects. Moreover, burst number was reduced since the third session of both administration conditions until the end of treatment. Interestingly, the effect of memantine of reducing the activation of ingestive behaviour was less pronounced in this study with respect to that observed with the previous study post-session administration schedule, in spite of the longer treatment. This apparent paradox might be explained if one considers these effects as instances of a memory-related effect, such as the development of CTA. In the framework of this hypothesis, the "before testing" sessions, not being followed by memantine administration, can be considered as extinction sessions performed every other day. Moreover, the animals treated with memantine at the highest dose failed to recover to pre-treatment ingestion levels 21 days after treatment discontinuation, while the animals treated after testing sessions in the previously published study showed a complete recovery well before the 15th day test. Within the same interpretative framework, this might depend by the reduced number and frequency of the extinction trials—i.e. the number of the sessions run after treatment discontinuation—in the present study. These results provide further support to the conclusion that memantine administration before sessions reduce burst size, an effect which is likely due to blockade of NMDA receptors occurring during behavioural testing. The observation that this effect can be obtained even in absence of a reduced intra-burst lick rate, which rules out the involvement of motor impairment, provides

Fund 2019 (PSD). The funders had no role in study design, data collection and analysis, decision to publish, or preparation of the manuscript.

**Competing interests:** The authors have declared that no competing interests exist.

an important piece of evidence in support to the interpretation of this effect as a blunted hedonic response. Moreover, these results provide further evidence that burst number reduction is due to a memory-related effect induced by memantine administration after sessions.

## Introduction

The uncompetitive N-methyl-D-aspartate (NMDA) receptor antagonist memantine—an "open channel blocker"–unlike other drugs of the same class such as phencyclidine and ketamine, is devoid of clinically unacceptable side effects such as dissociative symptoms and hallucinations [1–3]. Several preclinical [4–7] and clinical studies [8–11] support its use in the treatment of obesity and eating disorders, particularly binge-eating disorder. Memantine systemic administration reduced binge-like consumption of highly palatable food in rats [6, 7] and baboons [4, 5]. Acute administration was reported to reduce operant binge-like overeating of a highly palatable sugary reward, food seeking under a second order schedule of reinforcement and compulsive eating. These effects of memantine were selective for palatable food *versus* ordinary lab chow [7]. To account for these observations it was suggested that NMDA receptor blockade reduced the reinforcement related to the hedonic non-homeostatic mechanisms of food-intake [4–7]. Importantly, these effects were present after acute treatment, in the course of prolonged treatment, and after treatment discontinuation [6].

NMDA receptors are ubiquitously distributed in the mammalian CNS but show regional subunit composition [12]. In particular, NMDA-R2B receptor expression in the adult rat brain is limited to the forebrain [13]. Interestingly, it was reported that microinfusion of memantine in the nucleus accumbens shell—a limbic forebrain structure—reduced binge-like consumption of a highly palatable sugary reward [7].

Repeated exposure to sucrose reward—which depends both on taste and on caloric content —can induce neuroadaptations leading to the release of eating behaviour from the control of homeostatic mechanisms, thus resulting in compulsive overeating [14]. The analysis of the licking microstructure reveals specific responses to taste or hunger/satiety internal cues [15–20], which are related to the process by which reward evaluation regulates behavioural activation [15, 21–25]. In more detail, the number of lick bursts—i.e. discrete series of licks at the rate of about 5 to 7 licks *per* second [16]–indicates the number of times that the subjects engage in licking, and is mainly influenced by stimuli, such as post-ingestive cues, that do not involve orosensory contact with the drink solution [17, 19, 20]. In contrast, the size of the licking bursts, i.e. the number of licks *per* burst, is mainly dependent on the orosensory contact with the reward, and in particular by quality and intensity of taste of the drink solution [15–20]. Therefore, burst number and burst size might represent, respectively, (i) a process of "behavioural activation" (for a definition, see [26, 27]), and (ii) a process of reward evaluation, possibly related to the experience of pleasure [15, 22, 24, 25, 28].

To further elucidate the motivational aspects of the mechanisms by which memantine influences ingestive behaviour, two studies investigating memantine effect on the microstructure of licking for sucrose were performed in our laboratory. In order to evaluate the relationship between treatment effects and drug action at specific functionally relevant times, the effect of administration schedules involving different administration times in relation to the testing sessions was investigated. In a first experiment—which is reported in the present paper —a within-subject design was adopted. Memantine was administered in daily injections for 13

**Table 1. Experiment timeline.**

| Training | Treatment (until d13) | | | | | | | | | | | | | | Post-treatment |
|---|---|---|---|---|---|---|---|---|---|---|---|---|---|---|---|
| 8 daily sessions | d1 | d2 | d3 | d4 | d5 | d6 | d7 | d8 | d9 | d10 | d11 | d12 | d13 | d14 | **21 days** |
| | ↓S1 | S1'↓ | ↓S2 | S2'↓ | ↓S3 | S3'↓ | ↓S4 | S4'↓ | ↓S5 | S5'↓ | ↓S6 | S6'↓ | ↓S7 | S7' | 1st week: 6 sessions |
| | | | | | | | | | | | | | | | 14th day: 7th session |
| | | | | | | | | | | | | | | | 21st day: last session |

d: day; S: "before testing" session; S': "after testing" session; arrows indicate memantine injections either before or after experimental sessions (see Materials and methods for more details).

days, on alternate days, either 1-h before or immediately after a 30-min session in the same group of subjects (see Table 1 for a timeline of the experiment). Such a design results in seven "before testing" sessions—with memantine administered 1-h before sessions—and seven "after testing" sessions—with memantine administered immediately after the testing sessions. A similar protocol was adopted in a recent study investigating the effects of imipramine on ingestion [29]. Since the differences observed between the "before testing" and the "after testing" sessions could not be easily accounted for by the presumed different brain drug levels at testing time, to tease apart the effects related to the different administration time, a second experiment was performed adopting a between-subject design, investigating the effect of memantine daily treatment for seven days with the drug administered either 1-h before or immediately after the testing sessions in two separate groups of subjects [30].

"Before testing" and "after testing" sessions of the experiment reported in the present paper might be meaningfully compared to the testing sessions of the "before testing" and of the "after testing" groups, respectively, of the already published experiment [30], as far as the presumed memantine brain levels at testing time are concerned. As reported in a previous study, the brain maximal concentration ($C_{max}$) of memantine in rats following intra-peritoneal injection was reached after $68.5 \pm 3.4$ minutes ($T_{max}$), with a half-life of $2.8 \pm 0.5$ hours [31]. According to these data, the test sessions were run at the time of the brain $C_{max}$ when memantine was administered 1-h before tests, and with brain levels of less than 1% of $C_{max}$ when memantine was administered 23–25-h before tests, i.e. 1-h before the preceding session in the experiment reported in the present paper—or immediately after the preceding session in the previously published study. Therefore, only the effects observed in the "before testing" sessions of this study—and in the "before testing" group of the previously published study [30]–can be accounted for by NMDA receptor blockade (or by activity at other receptors) during testing sessions.

In the above cited study [30], daily treatment before testing sessions for seven days reduced ingestion due to reduced burst size, thus suggesting a blunted hedonic/reward evaluation response, an effect likely due to NMDA receptor blockade occurring during testing time. Notably, this effect was accompanied to reduced intra-burst lick rate, which leaves open the possibility of a role for motor impairment in the observed effect. On the other hand, daily post-session administration for seven days in a separate group of subjects induced a dramatic decrease of the activation of licking behaviour, indicated by reduced burst number. As a possible explanation for this response pattern, we hypothesised the development of a memory-related effect, such as conditioned taste aversion (CTA) [30] (see also [32–34]).

Although the between-subject design of the previously published experiment [30] is better suited to compare the effects of memantine administration before *versus* after testing sessions, the results of the within-subject study reported here provide additional evidence in support to

the proposed interpretation of the results. In particular, the present results show that burst size reduction induced by memantine can be accomplished even in absence of reduced intra-burst lick rate, thus strengthening the interpretation of the effect of pre-session memantine administration in terms of a blunted hedonic response. Moreover, the comparison of the results of the two experiments reveals (i) that the extent of the reduction of burst number does not depend on the length of treatment and (ii) that recovery after treatment discontinuation does not depend on time but on the number and frequency of the sessions performed. These observations provide further support to the interpretation of the effect of memantine post-session administration as a memory-related effect.

## Materials and methods

### Subjects and drug treatments

Experimentally naïve male Sprague-Dawley rats (Harlan, Italy) aged about 10 weeks and weighing 300–350 g at the beginning of the experiments were used as subjects. The animals were housed in groups of two-three per cage in controlled environmental conditions (temperature 22–24°C; humidity 50–60%; light on at 08:00, off at 20:00), with free access to food and water. Memantine HCl was purchased as an injectable commercially available pharmaceutical form (Ebixa, Lundbeck, Denmark) in ampoules containing an aqueous solution at the concentration of 10 mg/ml and, after dilution with distilled water when appropriate, was administered intraperitoneally (i.p.) at the doses of 10, 5 and 1 mg/kg, in a volume of 1 ml/kg. Vehicle treatment consisted in a 1 ml/kg distilled water i.p. administration.

### Ethics statement

All the experimental procedures were carried out in accordance with the regulatory requirement of the Italian law (D.L. 116, 1992) and Council Directive 2010/63EU of the European Parliament and Council, and were approved by the Istituto Superiore di Sanità (protocol n. 43890/2011) and authorised by the Ministry of Health, Italy (protocol n. 32/2012-B). At the end of the experiment, the rats were euthanised with pentobarbital sodium. Animals were monitored and properly handled throughout the experiment, and every effort was made to minimize suffering or pain.

### Apparatus, microstructural measures and testing conditions

Behavioural testing was carried out using a multistation lick analysis system (Habitest, Coulbourn Instruments, USA) connected to a computer. Rats were individually placed in a Perspex chamber with an opening in the centre of the front wall allowing access to a bottle spout. The recording period started either after the first lick or after 3-min that the animals were placed into the chambers, so that the latency to the first lick had a cut off time of 3-min. The interruptions of a photocell beam by each single tongue movement while licking the spout were recorded, with a temporal resolution to the nearest 20 milliseconds. The raw data were analysed through Graphic State 3.2 software (Coulbourn Instruments, USA) and, besides lick number, the following microstructural measures were obtained for each subject: number of bursts, time spent in bursts, latency to the first lick. A burst was defined as a series of licks with pauses no longer than 0.4-sec. Burst size (number of licks *per* burst) and intra-burst lick rate (lick/sec within bursts) were then calculated (see [21]). The experiments were performed between 09:00 and 13:00, i.e. during the light phase of the lighting cycle.

## Procedures

The timeline of the experiment is summarized in Table 1. The subjects (N = 36) were first familiarised with the test apparatus until they reached a stable baseline for burst size. This was accomplished in 8 daily 30-min training sessions. In the training sessions, as well as in the following experimental sessions, they had access to a 10% sucrose solution. Based on the mean burst size of the $8^{th}$ training session, the subjects were allocated into four matched groups (n = 9). Each group was treated for the 13 successive days with daily i.p. injections of either vehicle or one of three doses of memantine (1, 5, 10 mg/kg). The drug was administered on alternate days either 1-h before or immediately after a 30-min testing session. A $14^{th}$ experimental session was performed after the last treatment day. The sessions preceded by drug administration (odd number experimental days) were referred to as "before testing" sessions (S1-S7), while the sessions followed by drug administration, along with the $14^{th}$ session (even number experimental days) were referred to as "after testing" sessions (S1'-S7'). This treatment schedule implies that the experimental sessions were performed either 1 hour or 25 hours after drug administration. Moreover, the behaviour was observed for 21 days after treatment discontinuation. In particular, daily tests (with the exception of a weekend day) were performed in the $1^{st}$ week of treatment discontinuation. Two further tests were performed at days 14 and 21 of treatment discontinuation. Memantine doses were chosen taking into account previous studies examining the effect of memantine in binge-eating related animal models [6, 7].

## Statistical analysis

Statistical analysis of all sets of data was performed with ANOVA, by the software Statistica 8.0 (StatSoft Inc.). Post-hoc analysis of the main effects was made with the Newman-Keuls multiple comparison test. When a significant interaction between factors was revealed, comparisons were performed by F-test for contrasts.

Body weight data were analysed by ANOVA, with *dose* as a between-group factor and *time* as a within-group factor, with three levels corresponding to (i) the first treatment day, (ii) the last treatment day, and (iii) three weeks after the last treatment day.

ANOVA of the whole-session data involved the between-group factor *dose*, with 4 levels corresponding to memantine doses and the relative vehicle, and the within-group factor *session*, with 7 (or 8) levels, corresponding to treatment (or discontinuation) sessions. The analysis of the data relative to the treatment sessions involved a further within-group factor: *administration*, with two levels, *before testing* and *after testing*.

## Results

### Body weight

Body weight data are depicted in Fig 1. The comparison between the four groups matched according to the burst size of the last training session did not show any statistically significant difference (see Table 2, Matching). In the course of the experiment (from the first treatment day to the $21^{th}$ day after treatment discontinuation), body weight showed a steady increase in all groups (effect of *time*, see Table 2, All data). F-tests for contrasts based on the statistically significant *dose×time* interaction (see Table 2, All data) failed to show statistically significant differences in the comparisons between the groups treated with each of the 3 memantine doses and the vehicle-treated group at any time.

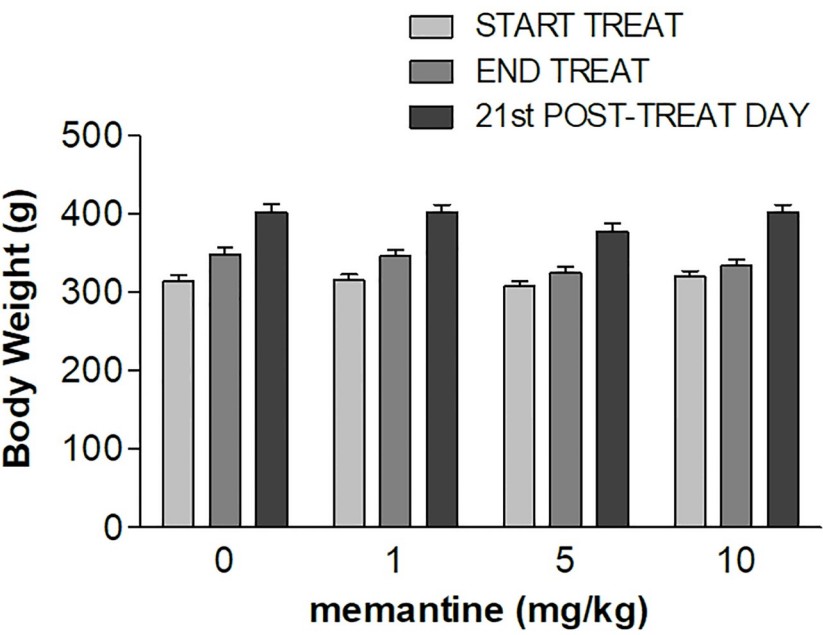

**Fig 1. Body weights.** TREAT: treatment. START TREAT: 1st treatment day. END TREAT: last treatment day. Values represent the mean ± S.E.M. from 9 subjects.

### Effect of treatment with memantine

**Lick number.** The comparisons between the groups treated with different memantine doses and the vehicle-treated group for the lick number data (Fig 2, top panels) are based on a *dose×session* interaction, since the three way interaction involving the factor *administration* was not significant (see Table 3), thus they refer to differences between doses regardless of administration time. Treatment with memantine at 10 and 5 mg/kg resulted in a reduction of lick number both in the "before testing" and in the "after testing" sessions. Such reduction was apparent since the first session of both administration conditions (S1 and S1') in the group treated with 10 mg/kg and since the second session of both administration conditions (S2 and S2') in the group treated with 5 mg/kg. A further decrease in the successive sessions was observed in these groups in both administration conditions. A statistically significant increase with respect to the vehicle-treated group was observed in the 6th session of both administration conditions (S6 and S6') with the dose of 1 mg/kg. Moreover, a significant *administration×dose* interaction was accounted for by a slight attenuation of the memantine-induced decrement of lick number in the "after testing" sessions.

**Table 2. Body weight data ANOVA.**

| *Factors* (df) | Matching | | All data | |
|---|---|---|---|---|
| | **F** | **P** | **F** | **P** |
| *dose* (3,32) | 0.56 | n.s. | 1.1 | n.s. |
| *time* (2,64) | – | – | 3,32 | n.s. |
| *dose×time* (6,64) | – | – | 4.54 | <0.001 |

df: degrees of freedom; n.s.: P>0.05, not significant.

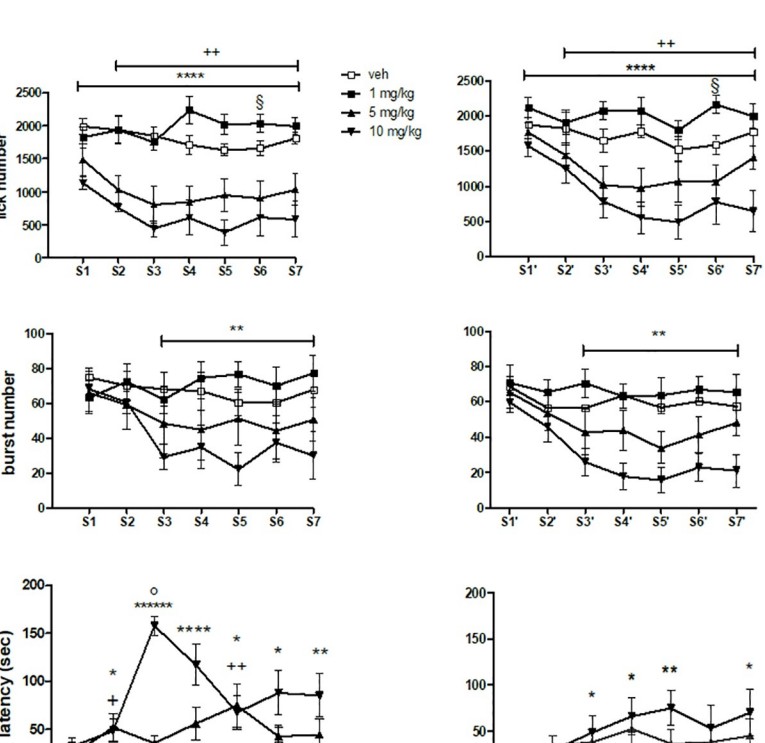

**Fig 2. Lick number, burst number and latency to the first lick during a 13 day daily treatment with memantine.**
Left and right panels report, respectively, the data from the "before testing" sessions and the "after testing" sessions. S: session. S1-S7 (left panels) were performed in odd number days and S1'-S7' (right panels) in even number days in a period of 14 days. Values represent the mean ± S.E.M. from 9 subjects. Memantine 10 mg/kg *versus* vehicle: *P<0.05, **P<0.01, ****P<0.00001, ******P<10^{-6}; memantine 5 mg/kg *versus* vehicle: +P<0.05, ++P<0.01; memantine 1 mg/kg *versus* vehicle: §P<0.05; "before testing" session (S3) *versus* "after testing" session (S3') in memantine 10 mg/kg: °P<10^{-6} (ANOVA followed by F-test for contrasts; straight lines indicate contrasts involving consecutive time points). Notice that the comparisons between the groups treated with different memantine doses and the vehicle-treated group for lick number (top panels) and burst number (mid panels) are based on a two way interaction (*dose×session*) not involving *administration*, thus they refer both to the "before testing" sessions and to the "after testing" sessions.

**Table 3. Treatment phase data ANOVA.**

| *Factors* (df) | LN | | BN | | LAT | | NLPB | | IBLR | |
|---|---|---|---|---|---|---|---|---|---|---|
| | F | P | F | P | F | P | F | P | F | P |
| *dose* (3,32) | 13.73 | <10^{-5} | 3.74 | <0.05 | 11.74 | <0.001 | 1.72 (3,23) | n.s. | 1.11 (3,23) | n.s. |
| *sess* (6,192) | 12.23 | <10^{-6} | 13.6 | <10^{-6} | 4.32 | <0.001 | 1.13 (6,138) | n.s. | 3.47 (6,138) | <0.01 |
| *adm* (1,32) | 12.33 | <0.01 | 5.72 | <0.05 | 9.71 | <0.01 | 38.3 (1,23) | <10^{-5} | 8.54 (1,23) | <0.01 |
| *dose×sess* (18,192) | 3.25 | <0.0001 | 4.16 | <10^{-6} | 4.19 | <10^{-6} | 1.2 (18,138) | n.s. | 1.45 (18,138) | n.s. |
| *dose×adm* (3,32) | 4.34 | <0.05 | 0.11 | n.s. | 4.42 | <0.05 | 6.68 (3,23) | <0.01 | 4.14 (3,23) | <0.05 |
| *sess×adm* (6,192) | 2.01 | n.s. | 1.4 | n.s. | 1.37 | n.s. | 0.62 (6,138) | n.s. | 2.39 (6,138) | <0.05 |
| *dose×sess×adm* (18,192) | 1.62 | n.s. | 1.19 | n.s. | 2.22 | <0.01 | 0.52 (18,138) | n.s. | 1.57 (18,138) | n.s. |

LN: lick number; BN: burst number; LAT: latency to first lick; NLPB: number of licks *per* burst; IBLR: intra-burst lick rate; sess: session; adm: administration; df: degrees of freedom; n.s.: P>0.05, not significant. Degrees of freedom relative to NLPB and IBLR data are indicated below F-values. The difference from the other data sets is due to empty cells (subjects with no licks in a session).

**Burst number.** The comparisons between the groups treated with different memantine doses and the vehicle-treated group for the burst number data (Fig 2, mid panels) are based on a *dose×session* interaction, since the three way interaction involving the factor *administration* was not significant (see Table 3), thus they refer to differences between doses regardless of administration time. Treatment with memantine at 10 mg/kg resulted in a reduction of burst number both in the "before testing" and in the "after testing" sessions since the third session of both administration conditions (S3 and S3'), with a further decrease in the successive sessions in both administration conditions. The significant effect of *administration* (with no significant interaction with *dose*) might be accounted for by the slightly reduced values in the "after testing" sessions regardless of either treatment group or session.

**Latency to the first lick.** F-tests for contrasts based on a *dose×session×administration* interaction (see Table 3) showed no differences between the groups treated with the different memantine doses both in the first "before testing" session and in the first "after testing" session. In the "before testing" sessions, higher latency values with respect to the vehicle-treated group were observed since the 2nd session (S2) up to the last one in the group treated with 10 mg/kg, with a peak in the 3rd session (S3). This was the only point showing a statistically significant difference with respect to the same group value in the corresponding "after testing" session (S3'). An increase in latency was also observed with the 5 mg/kg dose, with statistically significant differences in the third (S3) and in the fifth (S5) "before testing" sessions (Fig 2, bottom left). Higher latency values with respect to the vehicle-treated group were observed in the group treated with 10 mg/kg from the third (S3') to the last (S7') "after testing" session, with the exception of S6' (Fig 2, bottom right).

**Burst size: Number of licks per burst.** The comparisons between the groups treated with different memantine doses and the vehicle-treated group for the number of licks *per* burst data (Fig 3, top panels) are based on a *dose×administration* interaction, since the three way interaction involving the factor *session* was not significant (see Table 3), thus they refer to differences between doses regardless of session. A significant reduction of the number of licks *per* burst at the doses of 5 and 10 mg/kg was revealed in all the "before testing" sessions (Fig 3 top left panel), but not in the "after testing" sessions (Fig 3 top right panel).

**Intra-burst lick rate.** The intra-burst lick rate data are shown in Fig 3 (bottom panels). The results of the ANOVA are shown in Table 3. F-tests for contrasts based on the *dose×administration* interaction failed to reveal statistically significant differences. F-test for contrasts performed on the basis of the *session×administration* interaction showed a significant difference ($P < 10^{-5}$) between the 1st "before testing" session (S1) compared to the 1st "after testing" session (S1'), possibly due to the low values in the intra-burst lick rate in the group treated with memantine at 10 mg/kg in the "before testing" session.

## Effect of the discontinuation of treatment with memantine

**Lick number.** F-test for contrasts based on the *dose×session* interaction (see Table 4) showed significantly lower values in the group treated with memantine at 10 mg/kg, compared to the vehicle-treated group, from the first to the last session (21th day) after treatment discontinuation, with this difference progressively decreasing across sessions. Only in the session performed after 14 days of treatment discontinuation, no statistically significant differences between the memantine dose groups were observed (Fig 4, top left panel).

**Burst number.** F-test for contrasts based on a *dose×session* interaction (see Table 4) showed a significant decrease in burst number limited to the the 1st three sessions in the group treated with memantine at 10 mg/kg (Fig 4, mid left panel).

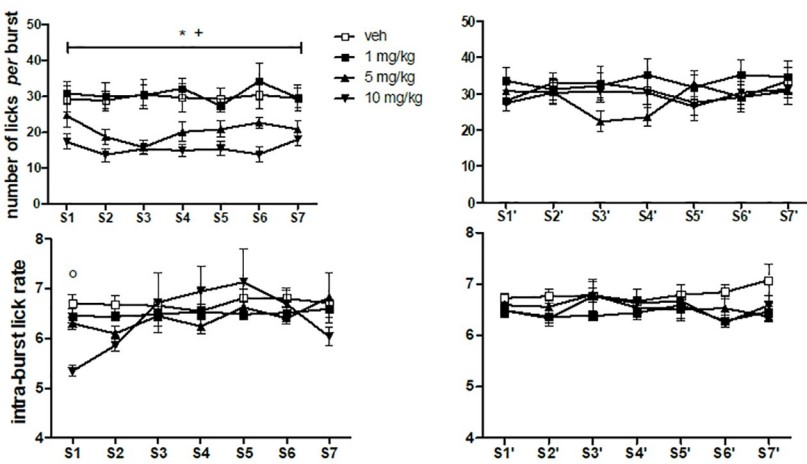

**Fig 3. Number of licks *per* burst and intra-burst lick rate during a 13 day daily treatment with memantine.** Left and right panels report, respectively, the data from the "before testing" sessions and the "after testing" sessions. S: session. S1-S7 (left panels) were performed in odd number days and S1'-S7' (right panels) in even number days in a period of 14 days. Values represent the mean ± S.E.M. from 9 subjects. Memantine 10 mg/kg *versus* vehicle: *P<0.05; memantine 5 mg/kg *versus* vehicle: +P<0.05; "before testing" session (S1) *versus* "after testing" session (S1') regardless of dose: °P<10⁻⁵ (ANOVA followed by F-test for contrasts; straight lines indicate comparisons involving consecutive time points). Notice that the comparisons between the groups treated with different memantine doses and the vehicle-treated group for number of licks *per* burst are based on a two way interaction (*dose×administration*) not involving *session*, thus they refer to all sessions within each administration time condition.

**Latency to the first lick, burst size (number of licks per burst) and intra-burst lick rate.** Comparisons (Newman-Keuls test) based on the main effect of *dose* (see Table 4) showed a statistically significant increase of latency values in the group treated with 10 mg/kg with respect to all the other groups (Fig 4, bottom left panel).

ANOVA of the number of lick *per* burst data (Fig 4, top right panel) failed to show any statistically significant effect or interaction (see Table 4).

ANOVA of the intra-burst lick rate data (Fig 4, bottom right panel) showed only a statistically significant effect of *session* (see Table 4) accounted for by slight differences between sessions regardless of treatment group.

## Discussion

To further explore the mechanisms underlying memantine effects on the motivational aspects of ingestive behaviour, the lick pattern in response to sucrose in the course of daily treatment

**Table 4. Post-treatment phase data ANOVA.**

| *Factors* (df) | LN | | BN | | LAT | | NLPB | | IBLR | |
|---|---|---|---|---|---|---|---|---|---|---|
| | F | P | F | P | F | P | F | P | F | P |
| *dose* (3,32) | 5.49 | <0.01 | 7.53 | <0.001 | 3.79 | <0.05 | 1.4 | n.s. | 1.55 | n.s. |
| *sess* (7,224) | 2.38 | <0.05 | 245 | <10⁻⁶ | 4.48 | <0.001 | 1.23 | n.s. | 2.08 | <0.05 |
| *dose×sess* (21,224) | 2.38 | <0.001 | 8.49 | <10⁻⁶ | 1.41 | n.s. | 0.79 | n.s. | 1.5 | n.s. |

LN: lick number; BN: burst number; LAT: latency to first lick; NLPB: number of licks *per* burst; IBLR: intra-burst lick rate; sess: session; df: degrees of freedom; n.s.: P>0.05, not significant.

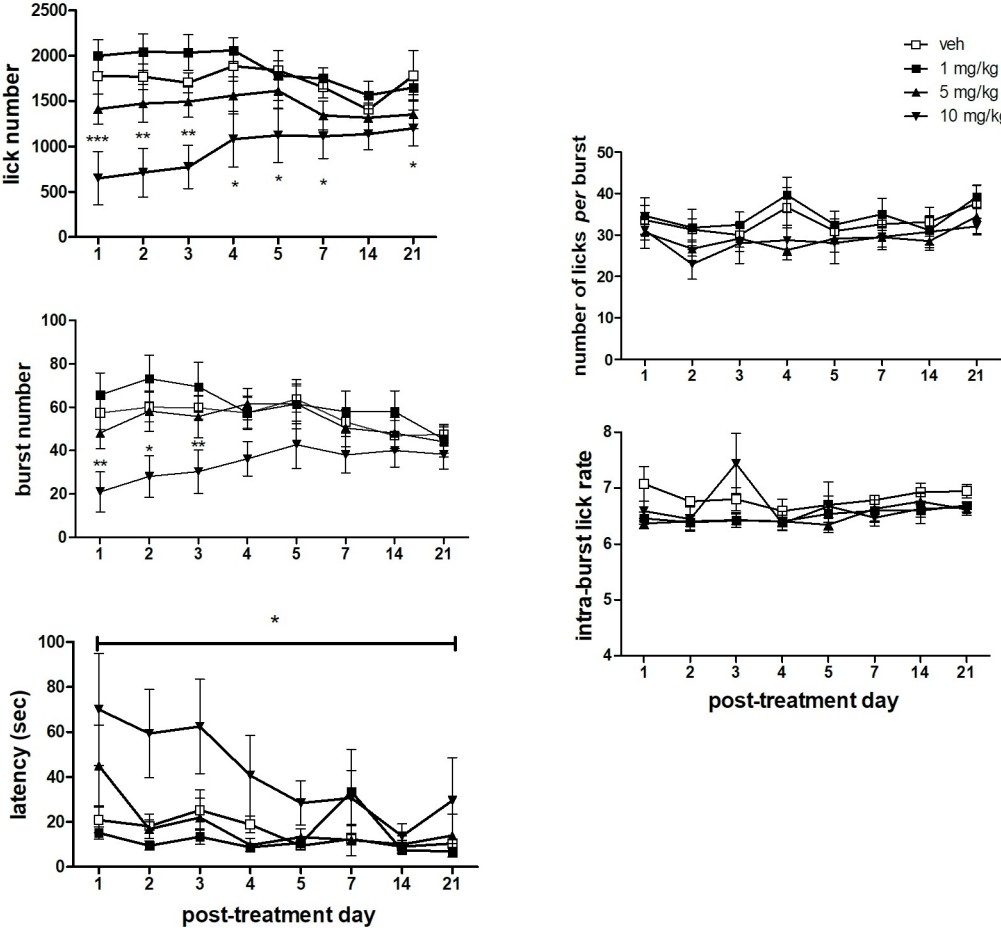

**Fig 4. Licking microstructural parameters after discontinuation of a 13 day daily treatment with memantine.**
Values represent the mean ± S.E.M. from 9 subjects. Memantine 10 mg/kg *versus* vehicle: *P<0.05, **P<0.01,
***P<0.001 (ANOVA followed by F-test for contrasts or Newman-Keuls test; straight lines indicate comparisons
involving consecutive time points).

and after treatment discontinuation was investigated in our laboratory. In a recently published
study [30], we compared the effect of memantine daily treatment for seven days with the drug
administered either 1-h before or immediately after the testing sessions in two separate groups
of subjects. In the present study, the combined effects of pre-testing and post-testing treatment
in the same group of subjects for thirteen days were investigated. Memantine was administered
on alternate days either 1-h before ("before testing" sessions) or immediately after ("after test-
ing" sessions) the daily licking tests.

In the first "before testing" session (S1), memantine at the dose of 10 mg/kg induced a
decrease of lick number (Fig 2, top left panel) exclusively due to reduced burst size (Fig 3, top
left panel), as observed in our previously published study in the group treated with memantine
1-h before testing, an effect which was interpreted as the consequence of a blunted hedonic
response [30].

The across-session time course of the effect of memantine at this dose in the successive
"before testing" sessions (S2-S7) consisted in a decreased lick number persisting up to the end
of the treatment phase (Fig 2, top left panel). In the "before testing" session S2 the reduced lick
number was accompanied only by a reduced burst size (Fig 3, top left panel). In all the

successive "before testing" sessions (S3-S7), also a decreased burst number was present (Fig 2, mid left panel). This response pattern was similar, but less pronounced, to the across-session reduction of activation of the licking response observed with the same memantine dose in the group treated immediately after testing in our previously published study, an effect which might be accounted for by a memory-related effect, such as CTA development [30]. In contrast, in the same study [30], burst number in the group of subjects treated with memantine 1 hour before sessions was unchanged after the first administration (as observed here), but showed a progressive increase in the following sessions. These observations, taken together, show that burst number reduction is induced by memantine post-session administration, and that, while given in combination, the effect of post-session administration on this parameter prevails over the effect of pre-session administration.

It might be worth noting that the peak in latency occurred concomitantly with the drop in burst number (S3, Fig 2, mid and bottom left panels). This might reflect the opposite relationships with the level of activation of licking behaviour, on the one hand, of the number of bursts —direct relationship—and, on the other hand, of the latency to first lick—inverse relationship [15, 30].

The same lick and burst number response pattern described in the "before testing" sessions was observed in the corresponding "after testing" sessions (Fig 2, top and mid right panels), but without effects on burst size (Fig 3, top right panel). Thus, burst size was reduced only in the "before testing" sessions, i.e. in the condition involving presumed pharmacologically relevant memantine brain levels. This suggests that this effect is due to an ongoing pharmacological action at testing time. Most importantly, at variance with the results of our previously published study [30], this effect was observed in absence of effects on the intra-burst lick rate (Fig 3, bottom left panel), which is an index of the motor competence necessary for licking [35–37]. This observation strengthens the interpretation of the effect of pre-session memantine as a blunted hedonic response.

Similar results were observed with the dose of 5 mg/kg, both in the "before testing" and in the "after testing" sessions, but the reduction in burst number was not statistically significant, while no effects were observed with the dose of 1 mg/kg, with the exception of an increased lick number limited to the sessions S6 and S6' (Figs 2 and 3). These observations suggest that the effect of memantine on sucrose licking is dose-dependent.

The observation that the burst number time-course of the "before testing" sessions and of the corresponding "after testing" sessions were virtually superimposable is consistent with the hypothesis that burst number reduction depends upon a memory-related effect on the level of activation of the licking response, such as the development of CTA. This hypothesis is also consistent with the observation that, in spite of a longer treatment duration, which implies a larger cumulative dose of memantine, the decrease of burst number observed in this experiment was less pronounced with respect to that observed in the group treated with memantine immediately after the testing sessions in our previously published study [30]. Indeed, in the framework of this hypothesis, the "before testing" sessions, not being followed by memantine administration, can be considered as extinction sessions performed every other day.

The results of the treatment discontinuation phase (Fig 4) showed a complete recovery of the effect of the 5 mg/kg memantine dose since the first session. As for the dose of 10 mg/kg, a significant reduction of lick number, accompanied by an increased latency time, persisted up to the last test, performed 21 days after treatment discontinuation. This result is in contrast with the results from our previously published study, which showed a complete recovery of all parameters well before the 15[th] day test [30]. One might be tempted to explain this difference as the result of the longer treatment—hence of the larger cumulative dose of memantine—in the present study. However, this account is difficult to reconcile with the observation discussed

above, i.e. that the reduced activation induced by memantine is less pronounced in this study with respect to that observed in the other study [30]. In the framework of the hypothesis that the detrimental effect of memantine treatment on the activation of licking behaviour is dependent on a memory-related effect—such as the possible acquisition of CTA–recovery should be dependent on the number of extinction trials, rather than on the mere passing of time. In keeping with this interpretation, after treatment discontinuation, eleven sessions were run in fifteen days in the previously published study, while in the present study the sessions were eight in twenty one days, i.e. the extinction trials were lower in number and frequency. Thus, these observations provide further support to the interpretation of the effect of memantine of reducing the activation of licking as a memory-related effect.

The results of a previous study showing the ability of acute administration of different NMDA receptor antagonists—namely phencyclidine and dizocilpine [38]–to reduce licking burst size, as observed here and in our previously published experiment in the sessions performed 1-h after memantine administration [30], strongly suggest that this effect is mediated by NMDA receptor blockade occurring during behavioural testing. However, the interpretation of this effect as a blunted hedonic response might be questioned by the observation that phencyclidine and dizocilpine [38]–as well as memantine in our previously published experiment [30]–reduced not only burst size but also the intra-burst lick rate, thus suggesting as an alternative interpretation the impairment of the motor competence necessary for licking [35–37]. As discussed above, the results of the present experiment show that the reduction of burst size induced by memantine can occur even in absence of effects on the intra-burst lick rate. In support of the notion that NMDA receptor antagonism results in a blunted hedonic response, dizocilpine was reported to reduce sucrose preference, mimicking the effect of sucrose dilution [39]. Finally, the interpretation of memantine effects in terms of a blunted hedonic response is consistent with previous findings showing the ability of memantine to reduce both appetitive and consummatory responses selectively for palatable foods [4–7].

Memantine treatment at all the doses tested failed to affect body weight gain, and no effects were observed after treatment discontinuation. This result is consistent with the results of a previous experiment performed in our laboratory in the same experimental conditions [30]–examining a dose of 10 mg/kg, and with the results of a previous study in a rat model of binge-eating disorder [6]–examining a dose of 5 mg/kg. Two uncontrolled open label studies in humans, both reporting reduced frequency of binge-eating episodes [8, 10], showed inconsistent results as for memantine effect on body weight, with only one study reporting a significant body weight reduction [10].

In conclusion, these results, consistently with the results of our previously published study [30], show that memantine administration before sessions results in reduced burst size, an effect which is likely due to blockade of NMDA receptors occurring during behavioural testing. In addition, the observation that this effect can be obtained even in absence of a reduced intra-burst lick rate, which rules out the involvement of motor impairment, provides an important piece of evidence in support to the interpretation of memantine effects on licking as a blunted hedonic response. This interpretation is consistent with the hypothesis that memantine interferes with the hedonic/non-homeostatic mechanisms regulating food-intake [4–7], and might be of relevance in accounting for the clinical effects observed in the treatment of binge-eating disorder [8–11]. Moreover, the difference in the reduction of the activation of licking behaviour between this experiment—combined effect of pre-session and post-session administration in the same subjects for 13 days—and our previously published experiment—in particular, effect of memantine administered after the testing sessions for seven days—can be easily explained if one considers these effects as instances of a memory-related effect,

such as the development of CTA. However, further experiments are necessary to test this hypothesis.

## Author Contributions

**Conceptualization:** Adriana Galistu, Paolo S. D'Aquila.

**Data curation:** Adriana Galistu.

**Formal analysis:** Adriana Galistu, Paolo S. D'Aquila.

**Funding acquisition:** Paolo S. D'Aquila.

**Investigation:** Adriana Galistu.

**Supervision:** Paolo S. D'Aquila.

**Writing – original draft:** Adriana Galistu, Paolo S. D'Aquila.

**Writing – review & editing:** Paolo S. D'Aquila.

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
