## [Decision Letter · Decision Letter 0]

13 Aug 2020

PONE-D-20-19768

Memantine effects on ingestion microstructure and the effect of administration time: A within-subject study

PLOS ONE

Dear Dr. D'Aquila,

Thank you for submitting your manuscript to PLOS ONE. After careful consideration, we feel that it has merit but does not fully meet PLOS ONE’s publication criteria as it currently stands. Therefore, we invite you to submit a revised version of the manuscript that addresses the points raised during the review process.

We look forward to receiving your revised manuscript.

Kind regards,

Alexandra Kavushansky, PhD

Academic Editor

PLOS ONE

- https://doi.org/10.1007/s00213-019-05348-3

In your revision ensure you cite all your sources (including your own works), and quote or rephrase any duplicated text outside the methods section. Further consideration is dependent on these concerns being addressed.

Additional Editor Comments (if provided):

Please address the issue of repeating an already published study. Please specifically highlight the differences between the current study and the one published previously, the rationale for the repetition and the novelty of the results.

Reviewers' comments:

Reviewer's Responses to Questions

**Comments to the Author**

1. Is the manuscript technically sound, and do the data support the conclusions?

Reviewer #1: Partly

Reviewer #2: No

2. Has the statistical analysis been performed appropriately and rigorously? 

Reviewer #1: Yes

Reviewer #2: N/A

3. Have the authors made all data underlying the findings in their manuscript fully available?

Reviewer #1: Yes

Reviewer #2: Yes

4. Is the manuscript presented in an intelligible fashion and written in standard English?

Reviewer #1: No

Reviewer #2: Yes

5. Review Comments to the Author

Reviewer #1: I appreciate the author presenting this research article emphasizing the effects of memantine on ingestion microstructure. My comments are as follows

1. Abstract is too long and not focus

2. Introduction: too long and not focus, also NMDA receptors especially NMDAR2B distribution in CNS should be

mentioned

3. The following comments should be explained

a. Why the body weight not decrease after menmantine tx? (Fig 1)

b Why administration of 1mg/kg is so different to 5mg/kg and 10 mg/kg (Fig.2) , even more improved?

c. Any complications seen in current study?

d. Lack of pathological finding to prove the hypothesis, at least cite the references

e. What are the limitation of current study?

Reviewer #2: The manuscript shows the results of a single experiment investigating the effects of memantine on sucrose microstructure.

Unfortunately, the authors have recently published a paper (Psychopharmacology (Berl)2020 Jan;237(1):103-114) which investigated the same topic, and this makes the results of this manuscript not novel.

6. PLOS authors have the option to publish the peer review history of their article (what does this mean?). If published, this will include your full peer review and any attached files.

Reviewer #1: No

Reviewer #2: No

---

## [Author Response · Author response to Decision Letter 0]

21 Aug 2020

Response to Reviewers, including the response to Journal Requirements and Additional Editor Comments

Our replies are in Italics. Please notice that in the Revised Manuscript with Track Changes the new text is highlighted with a light grey background, while the text overlapping with our previous publication (see below response to point 2 to Journal requirements) is highlighted with a cyan background. The line numbers reported in our responses refer to the Revised Manuscript with Track Changes.

and

Response: The article text was formatted according to the style requirements of PLoS One. Files were named according to the instructions reported in your letter. 

- https://doi.org/10.1007/s00213-019-05348-3

In your revision ensure you cite all your sources (including your own works), and quote or rephrase any duplicated text outside the methods section. Further consideration is dependent on these concerns being addressed.

Response: The text overlapping with the above linked publication (Galistu and D'Aquila, 2020, from our own lab) was rephrased. Please notice that several standard expressions were left unchanged in the revised text. The overlapping text is highlighted with a cyan background in the Revised Manuscript with Track Changes.

2. Additional Editor Comments:

Please address the issue of repeating an already published study. Please specifically highlight the differences between the current study and the one published previously, the rationale for the repetition and the novelty of the results.

Response: The main difference between the current study and the previously published study is in the design: a within-subject design in the current study, a between subject design in the previously published study. This differences is now clearly highlighted both in the abstract (see lines 16, 24) and in the Introduction (see lines 110, 118-119). The rationale of the study is reported in the Introduction (the text was modified, lines 107-109: “In order to evaluate the relationship between treatment effects and drug action at specific functionally relevant times, the effect of administration schedules involving different administration times in relation to the testing sessions was investigated”). In the revised version, it is cited a study from our laboratory, investigating the effects of imipramine on ingestion, adopting the same within-subject design to compare the effects of the drug at different administration times (lines 115-116). The additional findings with respect to the previously published study are highlighted in the last paragraph of the Introduction (lines 160-170) and in the Discussion (lines 426-430, 439-444, 448-456, 456-460, 469-470, 484-487, 492-496): (i) a reduced burst size 1-h after memantine administration can be observed even in absence of motor effects (ii) the strength of the effect of memantine post-session administration does not depend on the length of treatment (iii) recovery of ingestion to pre-treatment levels appears to be related to the number of sessions performed after treatment discontinuation rather than by discontinuation time. Point (i) provides further support to the interpretation of reduced burst size as a blunted hedonic response, points (ii) and (iii) provide further support to the interpretation of the effect of post-session administration as a memory-related effect. We hope that the changes to the text – aimed also to eliminate unnecessary information – might have increased the clarity of these points in the revised version.

3. Reviewer #1

I appreciate the author presenting this research article emphasizing the effects of memantine on ingestion microstructure. My comments are as follows

We thank the Reviewer for the appreciation of our work.

1. Abstract is too long and not focus

Response: The abstract text was shortened (from 492 to 439 words), with an effort to be more focussed. In particular, unnecessary information was deleted, while we tried to highlight the aspects which distinguish the present study from our previously published study on the same subject (please see also response to the Editor's comments above).

2. Introduction: too long and not focus, also NMDA receptors especially NMDAR2B distribution in CNS should be mentioned

Response: The Introduction was shortened (from 1253 to 1145 words) by eliminating unnecessary information. We appreciated the advice of the reviewer to mention NMDA receptor distribution. A short paragraph was added to deal with this point (lines 73-76).

3. The following comments should be explained

a. Why the body weight not decrease after menmantine tx? (Fig 1)

Response: A paragraph discussing the findings on body weight was added to the revised version (lines 475-481).

b Why administration of 1mg/kg is so different to 5mg/kg and 10 mg/kg (Fig.2) , even more improved?

Response: The difference between doses in memantine effect was dealt with in lines 431-435. The observation of the increased lick number with respect to vehicle – limited to sessions 6 and 6' – observed with the dose of 1 mg/kg was mentioned.

c. Any complications seen in current study?

Response: We could not identify any complications.

d. Lack of pathological finding to prove the hypothesis, at least cite the references

Response: Although the suggested interpretation of these results – in particular that memantine administered before sessions blunts the hedonic response to sucrose – might bear relevance in explaining the clinical effect of this drug in binge eating disorder, it does not have any implications in regard of the pathophysiology of eating disorders. Please notice that while some of the cited studies were aimed to model binge eating disorder (e.g. Popik et al., 2011, ref. 6; Smith et al., 2015, ref. 7), this is not the case for the present study. In the Introduction (line 60) and in the Discussion (line 480) we cited the few studies on the clinical effects of memantine on binge eating disorder.

e. What are the limitation of current study?

Response: We think that a limitation of this study might be the lack of conclusive evidence in regard to the interpretation of the reduced activation of ingestion due to post-session administration as CTA. This is acknowledged in the last sentence of the Discussion of the revised version.

Reviewer #2

The manuscript shows the results of a single experiment investigating the effects of memantine on sucrose microstructure.

Unfortunately, the authors have recently published a paper (Psychopharmacology (Berl)2020 Jan;237(1):103-114) which investigated the same topic, and this makes the results of this manuscript not novel.

Response: We are aware of the limited novelty of the findings reported in the present paper. In this regard it should be recalled that “PLOS ONE considers original research articles from all disciplines within science and medicine. The editors make decisions on submissions based on scientific rigor, regardless of novelty.” (https://www.editorialmanager.com/pone/default.aspx). Nonetheless, this issue was dealt with in the response to the Editor's comments, highlighting the relevant differences between this study and the previously published one, and the additional evidence provided by this study.

***

Finally, we would like to thank the Editor and the Reviewers for their comments and advice.

---

## [Decision Letter · Decision Letter 1]

3 Sep 2020

Memantine effects on ingestion microstructure and the effect of administration time: A within-subject study

PONE-D-20-19768R1

Dear Dr. D'Aquila,

We’re pleased to inform you that your manuscript has been judged scientifically suitable for publication and will be formally accepted for publication once it meets all outstanding technical requirements.

Kind regards,

Alexandra Kavushansky, PhD

Academic Editor

PLOS ONE

Additional Editor Comments (optional):

Reviewers' comments:

Reviewer's Responses to Questions

**Comments to the Author**

1. If the authors have adequately addressed your comments raised in a previous round of review and you feel that this manuscript is now acceptable for publication, you may indicate that here to bypass the “Comments to the Author” section, enter your conflict of interest statement in the “Confidential to Editor” section, and submit your "Accept" recommendation.

Reviewer #1: All comments have been addressed

2. Is the manuscript technically sound, and do the data support the conclusions?

Reviewer #1: Yes

3. Has the statistical analysis been performed appropriately and rigorously? 

Reviewer #1: Yes

4. Have the authors made all data underlying the findings in their manuscript fully available?

Reviewer #1: Yes

5. Is the manuscript presented in an intelligible fashion and written in standard English?

Reviewer #1: Yes

6. Review Comments to the Author

Reviewer #1: After full evaluation, I find the manuscript has been revised according to my comments item by item. Accept is my final decision

7. PLOS authors have the option to publish the peer review history of their article (what does this mean?). If published, this will include your full peer review and any attached files.

Reviewer #1: **Yes: **Jinn-Rung Kuo

---

## [Editor Report · Acceptance letter]

7 Sep 2020

PONE-D-20-19768R1 

Memantine effects on ingestion microstructure and the effect of administration time: A within-subject study 

Dear Dr. D'Aquila:

I'm pleased to inform you that your manuscript has been deemed suitable for publication in PLOS ONE. Congratulations! Your manuscript is now with our production department. 

Kind regards, 

on behalf of

Dr. Alexandra Kavushansky 

Academic Editor

PLOS ONE